# Therapeutic turnaround times for common laboratory tests in a tertiary hospital in Kenya

Thomas Mwogi[1,2,7]*, Tim Mercer[8], Dan N. (Tina) Tran[9], Ronald Tonui[3,4], Thorkild Tylleskar[1], Martin C. Were[5,6,7]

**1** Centre for International Health, University of Bergen, Bergen, Norway, **2** Directorate of Medicine, Moi Teaching and Referral Hospital, Eldoret, Uasin Gishu, Kenya, **3** Department of Immunology, Moi University, Eldoret, Uasin Gishu, Kenya, **4** Laboratory Services Division, Moi Teaching and Referral Hospital, Eldoret, Kenya, **5** Department of Biomedical Informatics and Medicine, Vanderbilt University Medical Center, Nashville, TN, United States of America, **6** Vanderbilt Institute for Global Health, Nashville, TN, United States of America, **7** Institute of Biomedical Informatics, Moi University, Eldoret, Uasin Gishu, Kenya, **8** Department of Population Health, The University of Texas at Austin Dell Medical School, Austin, Texas, United States of America, **9** Department of Pharmacy Practice, College of Pharmacy, Purdue University, West Lafayette, IN, United States of America

* thomasmwogi@mtrh.go.ke

**Data Availability Statement:** All relevant data are within the paper and its Supporting Information files.

## Abstract

Access to efficient laboratory services is critical to patient care. Turnaround Time (TAT) is one of the most important measures when judging the efficiency of any laboratory and care system. Few studies on TAT exist for inpatient care settings within low- and middle-income countries (LMICs).

### Methods

We evaluated therapeutic TAT for a tertiary hospital in Western Kenya, using a time-motion study focusing specifically on common hematology and biochemistry orders. The aim was to determine significant bottlenecks in diagnostic testing processes at the institution.

### Results

A total of 356 (155 hematology and 201 biochemistry) laboratory tests were fully tracked from the time of ordering to availability of results to care providers. The total therapeutic TAT for all tests was 21.5 ± 0.249 hours (95% CI). The therapeutic TAT for hematology was 20.3 ± 0.331 hours (95% CI) while that for biochemistry tests was 22.2 ± 0.346 hours (95% CI). Printing, sorting and dispatch of the printed results emerged as the most significant bottlenecks, accounting for up to 8 hours of delay (Hematology—8.3 ± 1.29 hours (95% CI), Biochemistry—8.5 ± 1.18 hours (95% CI)). Time of test orders affected TAT, with orders made early in the morning and those in the afternoon experiencing the most delays in TAT.

### Conclusion

Significant inefficiencies exist at multiple steps in the turnaround times for routine laboratory tests at a large referral hospital within an LMIC setting. Multiple opportunities exist to improve TAT and streamline processes around diagnostic testing in this and other similar settings.

**Funding:** This work was supported by the Norwegian Programme for Capacity Development in Higher Education and Research for Development (NORHED) (Norad: Project QZA-0484) and the Moi Teaching and Referral Hospital. The funders had no role in study design, data collection and analysis, decision to publish, or preparation of the manuscript.

**Competing interests:** The authors have declared that no competing interests exist.

## Introduction

A functional and accessible clinical laboratory infrastructure plays a crucial role in determining the diagnosis and treatment of communicable and non-communicable diseases alike. [1] Literature has shown the importance of clinical laboratories in facilitating clinical decision-making processes in a range of clinical diseases. [2–4] Inadequate access to quality-assured laboratory results often leads to further wastage of limited resources and potential harm to patients. [5]

Access to well-equipped diagnostic testing is limited in low- and middle-income countries (LMICs), especially in Sub-Saharan Africa (SSA). [6, 7] Barriers to reliable laboratory testing include: inadequate health-care infrastructure to support laboratory capacity, poor quality of laboratory facilities, low availability of equipment and supplies, lack of implementation of standardized operating procedures, and lack of adequate personnel. Without diagnostic testing support, misdiagnosis (i.e. under- or over-diagnosis) based on clinical signs and symptoms occur frequently. [7] While improving access to diagnostic equipment and laboratory resources represents a crucial step to improving health outcomes in LMICs, other opportunities exist to ensure that quality diagnostic testing is done in a timely, cost-effective and efficient manner. [8, 9]

An opportunity to improve diagnostic testing relies on identification of laboratory workflow to identify bottlenecks in turnaround time (TAT). Workflow evaluation helps in rethinking of processes and can help clinical laboratories do more with less. [10] Improving workflow efficiency in the laboratory is a cost-effective approach to maximizing health benefits for patients despite limited resources being available. [10] Quality improvement efforts geared towards improving the workflow have shown improved efficiency in hospital care settings within LMICs. [11–13] Human error, communication system breakdowns, redundant work steps and slow TAT all contribute to reduced workflow efficiency. [14] Redundant steps in the laboratory testing workflow are particularly common in LMIC settings, which commonly use paper-based laboratory service requests and results reporting. Such redundancies include filling entry and exit logs, and signing in and out samples and laboratory results. Lost paper requests and laboratory reports often mean another set of documentation. All these deficiencies further increase the TAT for results, with poor health consequences downstream. [15–17]

Quality has been defined as the ability of a service or product to satisfy the needs of a customer. [18] Clinical laboratories have traditionally focused on imprecision and inaccuracy to define the quality of results. This is a restrictive definition that focuses only on the technical aspect. Comprehensive laboratory result quality to the clinician encompasses precision, accuracy, availability, cost, relevance and timeliness. [19] Timeliness is considered to be one of the most crucial measures as it has a significant impact on patient care and satisfaction. It is for this reason that we are seeing increased use of point of care (POC) testing instruments. [19] Timeliness for diagnostic testing is commonly measured using the TAT.

To various stakeholders, TAT is often variably defined. However, for care providers who order the tests, TAT has most relevance in its definition as the time from the ordering of the test to the time when the result is available to the clinician. Delays relating to pre and post analytic phases are estimated to be responsible for up to 96% of total TAT, and a simple intra-laboratory definition of TAT risks grossly underestimating clinically-relevant TAT. [20]

Lundberg first outlined the activities involved in the performance of a laboratory test as a series of nine steps, namely: ordering, collection, identification, transportation, preparation, analysis, reporting, interpretation and action. [21] He defined the TAT that involves all the nine steps as the brain to brain TAT or the "therapeutic TAT". [21] The therapeutic TAT is the most comprehensive measure of timeliness of a clinical laboratory.

In LMICs like in other settings, many tests are ordered with need for timely access to results to help with critical care decisions. The aim of this study was to measure the therapeutic TAT

for common hematological and biochemical analyses at a national referral hospital and to identify processes and factors that contribute most to delays in TAT.

## Methods

### Setting

This was a prospective, descriptive, single-center study of therapeutic TAT for common laboratory tests at a tertiary hospital in Kenya. The hospital has achieved ISO accreditation in Quality Management Systems (ISO 9001:2015 Standard) and Medical Laboratory Standard (ISO 15189:2012 Standard). The study took place at the adult medicine wards of the hospital. The hospital has 11 clinical laboratories, namely: hematology, biochemistry, microbiology, tuberculosis, immunology, histology/pathology, parasitology, blood bank, blood transfusion unit, private wing and children's unit. The laboratories operate 24/7 and testing is run continuously although samples are received in batches. There are no point of care tests done at the hematology and biochemistry laboratories although some point of care tests occur at the wards e.g. random blood sugar tests. These point of care tests were excluded from this study. None of the laboratories had a laboratory information system (LIS) at the time of the study. The laboratories operate daily with a total of 6 pathologists (1 full-time and 5 part-time) and 146 laboratory technologists under employment.

### Quantifying turnaround time

The Lundberg definition of TAT was used in this paper. [21] This means that the pre-analytical TAT used was from the point of order of tests to the receipt of samples at the laboratory. Similarly, the post-analytic phase started from the time results were available at the laboratory to the point where clinicians could access it for action. The Therapuetic TAT was quantified using a time motion analysis approach. During the study period, a trained research assistant (RAs) rounded daily with the Inpatient ward team, and followed the relevant laboratory tests ordered throughout all processing steps over a 24-hour period. Over a seven-week period, RAs tracked the two most commonly ordered tests, namely full hemogram (FHG) hematology tests and the Urea, Electrolyte and Creatinine (UEC) biochemistry tests—hematology and biochemistry tests are used subsequently to describe these tests in this paper. The first week of data collection was discarded to compensate for the Hawthorne effect, as clinician or laboratory staff behavior might change when they were initially being observed. [22] Standardized data collection forms were developed for data entry using REDCap tool (Figs 1–4), and these were loaded onto mobile devices for use by the RAs. During rounds, the RAs equipped with a mobile device with the REDCap data collection tool, recorded the time of test order, and then followed those tests throughout the laboratory workflow process, assigning a time stamp at each of the steps outlined below (Fig 5). The RAs also collected relevant laboratory time-stamps from the laboratory computers system, as time-stamps were generated when the laboratory test was both analyzed and when the results were printed.

In addition to laboratory workflow time-stamp data, we also collected other data to help in further evaluating the TAT times observed. These additional data collected included: number of clinicians, laboratory personnel, nurses and phlebotomists present at the time each laboratory test was being tracked. We also documented challenges noted by RAs during tracking of laboratory tests using a standardized coded list of items. Coded list of challenges included: (1) Misplaced laboratory order inaccessible to phlebotomist, (2) Patient declined sample to be taken, (3) Patient unavailable for sample collection, (4) Sample collected but misplaced before leaving the ward, (5) Sample left ward but was not received in the laboratory, (6) Sample was clotted, (7) Sample volume was insufficient for analysis, (8) Sample received in laboratory but

## LUNDBERG'S 9 STEP WORKFLOW

| PRE-ANALYTIC | ANALYTIC | POST-ANALYTIC |
| --- | --- | --- |
| Bedside, Inpatient Wards, Transportation | Biochemistry and hematology laboratories | Transportation, inpatient wards, nurse's desk |

**PRE-ANALYTIC**
- Order
- Identification
- Collection
- Transit to laboratory
- Accession & preparation

**ANALYTIC**
- Analysis

**POST-ANALYTIC**
- Result Reporting
- Interpretation
- Action

**Fig 1. Laboratory workflow process.** The figure shows the complete brain to brain workflow processes involved between the order of the common laboratory tests and the availability of the results to clinicians.

misplaced, (9) Analysis was done but result was not printed, and (10) Result printed but then misplaced.

Patient-level data with patient-identifiers were also collected temporarily to track the TAT of labs during a 24-hour period. The RAs needed to collect the patient's name, identification number, and location on the ward in order to locate the patient's laboratory test. These patient-level data were stored securely on a password-protected device. At the end of each 24-hour period, the patient-level data and protected health information were permanently destroyed. The study was approved by the Institutional Review and Ethics Committee at Moi University.

## Sample size determination

For tests with long TAT as was expected in the setting of this study, sample sizes between 100 and 500 are recommended in order to give reproducible means for TAT. Given that the

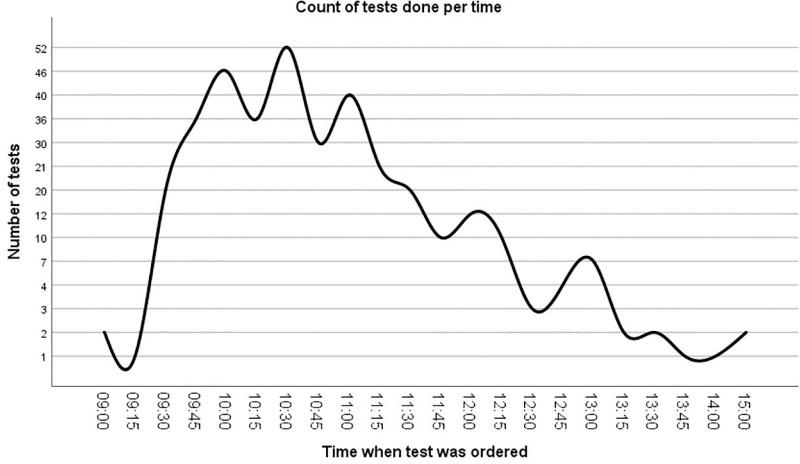

**Fig 2. Tests per time.** The figure shows the number of tests done per time of the day.

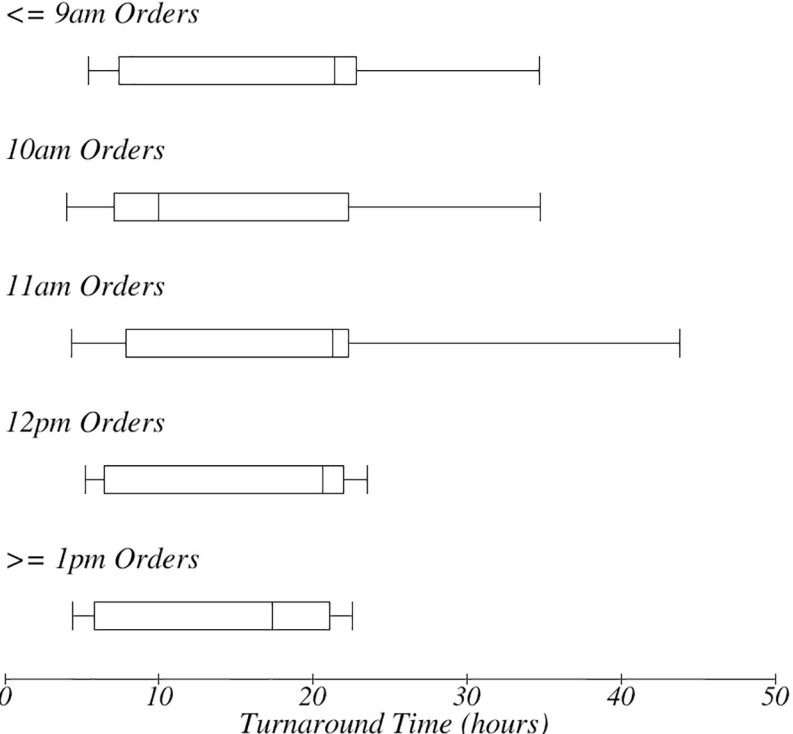

**Fig 3. TAT for time of order.** The box-plot shows the time the test was ordered and impact on TAT.

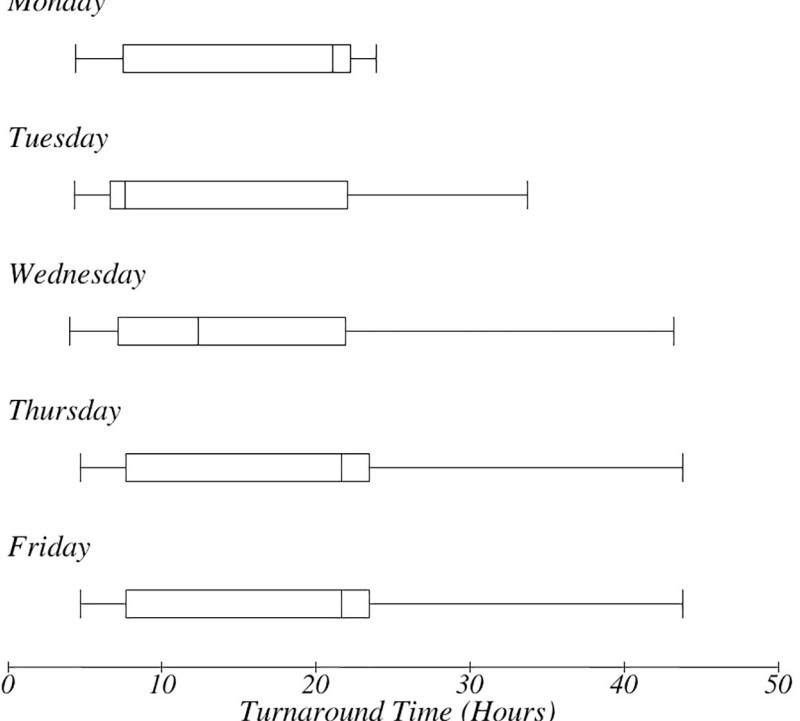

**Fig 4. TAT for day of order.** The box-plot shows the day of the week when the test was ordered and impact on TAT.

inpatient units chosen for this study typically sent orders for around 15 to 20 samples each of hematology or biochemistry tests per day, we chose a consecutive sampling approach. [14]

## Data analysis

The collected study data were extracted from the REDCap database and patient identifying information were removed as outlined above. Study personnel scanned these data for any inconsistencies in timestamps recorded, missing or invalid timestamps. The data were transferred to Microsoft Excel spreadsheets, with one spreadsheet each for hematology and biochemistry tests. The timestamps were then separated in columns in keeping with the Lundbergs nine-step workflow as shown in (Fig 5). [21] For each laboratory test record, time difference between one step and the subsequent step was calculated and recorded. For each time difference for every step, three calculations of TAT were done: Mean, Median and 90% completion time. These three measures are among four recommended by Steindel and Novis as being adequate and comprehensive measures of TAT. [23]

Given the long therapeutic TAT, the mean was used primarily as it is regarded to be a more objective measure in long TAT. This is based on a recommendation by Hawkins et al. [14] Boxplots were used to show the relationships of the TAT and the number of personnel present during the workflow process.

# Results

## Overall

A total of 460 laboratory tests (200 hematology and 260 biochemistry) belonging to 239 unique patients were tracked during a seven-week period between July and September 2018. To minimize the Hawthorne effect, the 42 laboratory tests that were tracked in the first week of the study were not included in the final analysis. Of the remaining 418 (180 hematology and 238 biochemistry) laboratory tests that were tracked, 62 (13.5%) were not fully processed and these included 26 (5.7%) hematology and 36, (7.8%) biochemistry), with the results never making it

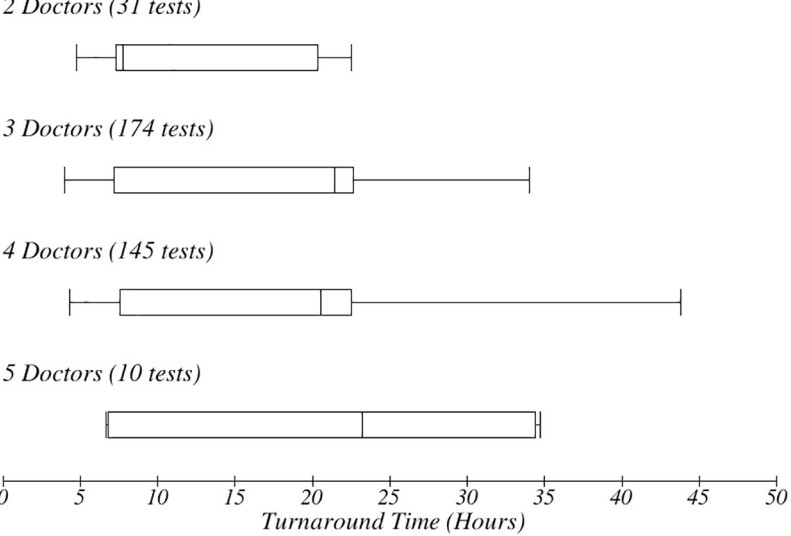

**Fig 5. Number of doctors and impact on TAT.** The box-plot shows the number of doctors present when the test was ordered and impact on TAT.

back to the clinical team that ordered the test. Reasons leading to non-completion of the tests are outlined in Table 1.

## Therapeutic TAT

The remaining 356 out of 418 tests (85.2%), made of 155 hematology and 201 electrolyte tests, went through the whole work-flow process. Table 2 summarizes the therapeutic TAT for hematology and electrolyte tests. The average therapeutic TAT for hematology was 20.3 ± 0.331 hours (95% CI) while that for biochemistry was 22.2 ± 0.346 hours (95% CI). The processing step that caused the biggest delay in TAT was 'Printing, sorting and dispatch' of results. The mean time taken by this step for hematology and biochemistry was 8.3 ± 1.29 hours (95% CI) and 8.5 ± 1.18 hours (95% CI) respectively. In both cases, the distribution was heavily skewed to the left resulting in the large standard deviation observed. Transportation of samples was the most efficient process with the mean transportation time for both hematology and biochemistry tests being around 10 minutes. Analysis took significantly longer with biochemistry when compared with hematology, which was the primary contributor to the significantly longer TAT overall for biochemistry compared with hematology tests (Table 2).

A majority of orders were done between 9:30am and 12:00pm with a peak at 10:30am (Fig 6). Orders done during peak times experienced the longest TAT (Fig 7). Orders done in the morning hours (up to 11am) experienced longer TAT when compared with orders done after 12pm (Fig 7).

There was a variation in the TAT based on the day of the week in which orders were made (Fig 8). Orders for tests done later in the week had longer TAT. There was no apparent relationship between the number of personnel present during the workflow process and the therapeutic TAT (Fig 9). However, it emerged that the more personnel were present, the longer the TAT (Fig 10). The more the number of orders, the more the number of personnel that were deployed to process the orders as well.

## Discussion

In essence, our study demonstrated that test results that had been analysed and were available for care could not be accessed by clinicians for another 18 hours. Our findings further add to the evidence that pre-analytical and post-analytical phases of laboratory processing contribute up to 96% of total TAT. [24]

**Table 1. List of tests misplaced in the workflow.**

| Reason for the test not getting to the clinician | Hematology | Biochemistry | Total |
|---|---|---|---|
| | # | # | # (% of Total) |
| Misplaced laboratory order inaccessible to phelobotomist | 3 | 4 | 7 (1.5) |
| Patient declined sample to be taken | 0 | 1 | 1 (0.2) |
| Patient unavailable for sample collection | 4 | 3 | 7 (1.5) |
| Sample collected but misplaced before leaving the ward | 2 | 5 | 7 (1.5) |
| Sample left ward but was not received in the laboratory | 4 | 6 | 19 (4.1) |
| Sample was clotted | 3 | 2 | 2 (0.4) |
| Sample volume was insufficient for analysis | 0 | 1 | 1 (0.2) |
| Sample received in laboratory then misplaced | 2 | 2 | 4 (0.9) |
| Analysis was done but result was not printed | 5 | 5 | 10 (2.2) |
| Result printed but then misplaced | 3 | 7 | 10 (2.2) |
| **Total** | **26** | **36** | **62 (13.5)** |

**Table 2. Turnaround time for specific time intervals of the workflow process.**

| | Hematology | | | Biochemistry | | | |
|---|---|---|---|---|---|---|---|
| | Mean | Median | 90th Perc | Mean | Median | 90th Perc | p-value |
| | Hrs (SD) | Hrs | Hrs | Hrs (SD) | Hrs | Hrs | (Mean) |
| Order to Sample collection | 2.18 (2.2) | 2.08 | 3.3 | 2.1 (1.2) | 2.15 | 3.24 | 0.6616 |
| Sample collection to transport | 1.24 (0.7) | 1.16 | 2.19 | 1.25 (0.7) | 1.17 | 2.17 | 0.8938 |
| Transport to received in laboratory | 0.15 (0.2) | 0.08 | 0.42 | 0.16 (0.3) | 0.1 | 0.42 | 0.7205 |
| Pre-analytic period | 0.71 (0.5) | 0.63 | 1.24 | 0.72 (1.6) | 0.5 | 1.22 | 0.9402 |
| Analysis | 1.06 (2.1) | 0.85 | 1.73 | 2.06 (2.5) | 1.55 | 3.17 | 0.0001 |
| Printing sorting and dispatch | 8.25 (8.2) | 2.33 | 17.36 | 8.47 (7.9) | 2.55 | 16.25 | 0.7979 |
| Transport to received in the ward | 2.30 (5.4) | 0.22 | 15.07 | 2.07(6.3) | 0.17 | 9.87 | 0.7167 |
| Received in ward to access by clinician | 7.99 (8.0) | 1.67 | 16.17 | 7.46 (8.8) | 1 | 16.2 | 0.5583 |
| Overall turnaround Time | 20.3 (2.1) | 9.02 | | 22.2 (2.5) | 9.19 | | 0.0001 |

Steindel and Novis identified four measures that can be used to adequately represent TAT. [23] These are the mean, median, 90th percentile and proportion of acceptable tests or outliers. In this study we used a combination of the mean, median and the 90th percentile in order to capture a comprehensive picture of TAT (Table 2).

Consolidated data available through external quality control programs like the CAP robes and Q-Track remain the reference point for laboratory TAT. A 2001 Q-Probes study concluded that the optimal time from order to reporting for biochemistry tests was 47 minutes while that for hematology was 35 minutes. [25] While comparisons with other studies is difficult because of varied definitions of TAT, it is still clear that the TAT in our study was significantly prolonged compared to recommended TAT for the tracked tests.

A time motion study done at the John Radcliffe Hospital (JRH), Oxford, UK in comparison determine that the TAT for hematology results was 1 hour 6 minutes (95% CI: 29 minutes to 2 hours 13 minutes) and that for biochemistry was 1 hour 42 minutes (95% CI: 1 hour 1 minute to 4 hours 21 minutes). [26] This was in a setting where result were immediately available to

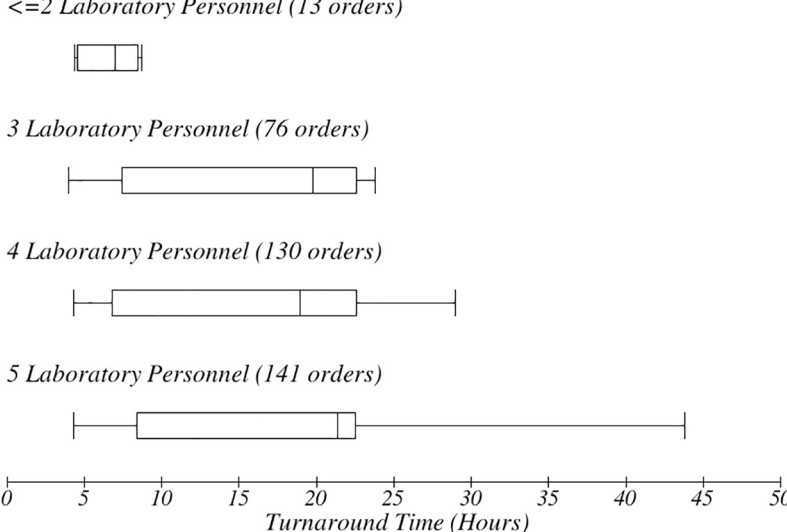

**Fig 6. TAT for time of order.** The box-plot shows the number of laboratory personnel present when the test was processed and impact on TAT.

Confidential

MTRH Laboratory Workflow Study

## Ward Lab Order Instrument

Record ID

____________________

Time of Record Creation

____________________

Record Creation Latitude

____________________

Record Creation Longitude

____________________

Record Created By

____________________

Patient ID

____________________

Patient Name

____________________

Select Lab Test
○ Full Hemogram
○ Urea Electrolytes & Creatinine

Time of Order

____________________
(Time when lab request is written)

Number of doctors in ward round

____________________

Ordering issues/problems

____________________
(Document any issues experienced that may have delayed lab orders - e.g. not enough personnel, no order papers etc)

**Fig 7. Ward to laboratory order instrument.** Instrument used to record initial laboratory order.

Confidential

MTRH Laboratory Workflow Study

## Sample Collection Instrument

**Patient ID: [patient_id]**
**Patient: [patient_name]**
**Lab Test: [select_lab_test]**

Record Updated By

____________________

Collection Latitude

____________________

Collection Longitude

____________________

Time of collection

____________________
(Time for identification and collection of sample)

Time Sample Leaves Ward

____________________

Number of Phlebotomists

____________________
(Number of sample collectors present in the ward at this given time)

Collection Issues

____________________

**Fig 8. Sample collection instrument.** Instrument used to record data during sample collection.

**Fig 9. Intra laboratory instrument.** Instrument used to record data during the analytical stage.

**Fig 10. Post laboratory instrument.** Instrument used to record data during the post analytical phase.

clinicians after analysis through electronic medical record systems. These and other observations demonstrate the importance of automating laboratory result delivery process, a step that was missing in our study setting as all steps were manual and paper-based.

In our study setting, printed results were batched for as long as 8.2 hours and 7.9 hours for CBC and UECs respectively. Printing and paper result delivery is not necessary in setups where computerized provider order entry (CPOE), laboratory information systems (LIS) and electronic medical record (EMR) systems have been implemented. Studies have shown that EMR and CPOE systems reduces both intra-laboratory and total TAT. [27] The impact of CPOE in our setting will probably be more significant given that up to 30% of analyzed results got misplaced—with half of the misplaced results being those that were analyzed but not printed, while the other half were printed and the paper result were untraceable.

Steindel and Novis suggest that 30 minutes as a reasonable time pre-analytic time, within which laboratory results should be received and verified. [23] In our study, for both hematology and biochemistry tests, the pre-analytic period lasted more than 30 minutes. This long pre-analytic time was partly a result of the manual recording processes needed to detail ordered tests in a paper register before being verified and received for processing.

Batching of the orders, of the collected samples and of results also contributed to the long overall TAT. Orders made early in the morning had a longer TAT as they were batched and had to wait for all orders before phlebotomy began (Fig 7). Orders done later in the day missed the batch and sometime could not get to the laboratory in time for analysis with the days' earlier batch. Pneumatic transportation systems eliminate the need for batching and ensure consistently low TAT regardless of the time of order. [28] In the study setting, pneumatic transport systems, and use of point of care tests where relevant, could serve to reduce increased TAT related to batching. [2, 9, 29]

It was observed that orders done later in the week took significantly longer to process. This may be associated with increased numbers of samples that needed to be processed as the week progressed (Fig 8).

It surprisingly emerged that when more personnel were present during the processing of orders, the overall TAT was longer (Figs 9 and 10). However, this could simply be a reflection of the fact that more personnel are deployed in times of crisis or when the ward is busiest, when TAT was already longer. This is an interesting finding that may need further exploration.

Limitations of our study include the fact that it was done within one referral hospital setting that might not be reflecting of other clinical settings even within other LMICs. Further, our assessment only involved hematology and biochemistry tests which were also all handled within the facility. TAT will likely be different for other tests and for send out tests. However, through this study, we provide a clear demonstration of the need to analyze TAT systematically within clinical settings in LMICs, and to implement mechanisms to mitigate long TAT. Another limitation of the study is that it only considered printed laboratory results. In cases where critical results were communicated either verbally or via text message, the study may overestimate TAT. However the number of communicated critical results are low in this setting.

There is need to streamline the steps involved in delivery of common laboratory results in the tertiary hospital. As the next step, we hope to implement technology-based solutions to help in quick processing of orders using computerized order entry approaches, and interfaces that allow results to be availed immediately to providers. Such a solution will have to be tailored for resource-limited settings that might have limited technological infrastructure and financial resources. An approach that uses a mobile-based solution tethered to laboratory information system could help address many of these challenges, and also help with timely data collection to ensure real-time tracking of deficiencies in TAT.

## Conclusion

This time motion study in a tertiary hospital in Kenya demonstrated that there are significant delays in delivery of hematology and biochemistry test results to clinicians in time. Despite efficient analysis of results, the post analytic period contributed the most delay resulting in more than 20 hours of therapeutic TAT. Printing, sorting and dispatch of results emerged as the greatest bottleneck in the process. Transportation was the most efficient process but this was in the context of batching of results before transport. There was a biphasic elongation of TAT with early morning and afternoon orders bearing the most delay. The results of this study elucidate specific bottlenecks and targets for interventions that could improve the efficiency of the laboratory workflow process ultimately improving clinical care.

## Acknowledgments

We acknowledge the immense support received from the hospital management and more specifically by the Chief Executive Officer, the head of the department of laboratory services Ms. Florence Tum and the deputy of the same department Mr. Philemon Chebii. We thank the research assistants: Carolyne Songok, Millicent Tanui and Olympia Cheruiyot for their dedication and attention to detail as well as going beyond their call of duty to ensure work done was as perfect as humanly possible.

## Author Contributions

**Conceptualization:** Thomas Mwogi, Tim Mercer, Dan N. (Tina) Tran, Ronald Tonui, Thorkild Tylleskar, Martin C. Were.

**Data curation:** Thomas Mwogi, Tim Mercer, Ronald Tonui, Thorkild Tylleskar, Martin C. Were.

**Formal analysis:** Thomas Mwogi, Thorkild Tylleskar, Martin C. Were.

**Funding acquisition:** Thorkild Tylleskar, Martin C. Were.

**Investigation:** Thomas Mwogi, Tim Mercer, Dan N. (Tina) Tran, Ronald Tonui.

**Methodology:** Thomas Mwogi, Tim Mercer, Dan N. (Tina) Tran, Ronald Tonui, Thorkild Tylleskar, Martin C. Were.

**Project administration:** Thomas Mwogi, Dan N. (Tina) Tran, Ronald Tonui, Martin C. Were.

**Resources:** Thomas Mwogi, Dan N. (Tina) Tran, Ronald Tonui, Thorkild Tylleskar, Martin C. Were.

**Software:** Thomas Mwogi.

**Supervision:** Thomas Mwogi, Ronald Tonui, Thorkild Tylleskar, Martin C. Were.

**Validation:** Thomas Mwogi, Tim Mercer, Thorkild Tylleskar, Martin C. Were.

**Visualization:** Thomas Mwogi.

**Writing – original draft:** Thomas Mwogi, Thorkild Tylleskar, Martin C. Were.

**Writing – review & editing:** Thomas Mwogi, Tim Mercer, Dan N. (Tina) Tran, Ronald Tonui, Thorkild Tylleskar, Martin C. Were.

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
