## [Decision Letter · Decision Letter 0]

19 Nov 2019

PONE-D-19-21599

Therapeutic turnaround times for common laboratory tests in a tertiary hospital in Kenya

PLOS ONE

Dear Dr Mwogi,

Thank you for submitting your manuscript to PLOS ONE. After careful consideration based on three expert reviewers we think that it has merit but does not fully meet PLOS ONE’s publication criteria as it currently stands. Therefore, we invite you to submit a revised version of the manuscript that addresses all the points raised during the review process.

Please make sure that you provide a detailed response to each comment. Your response should include the revised text as well as the location of the revision (page number and line numbers) in the text where that specific change can be found. This will facilitate the re-review of your manuscript.

We would appreciate receiving your revised manuscript by Jan 03 2020 11:59PM. To enhance the reproducibility of your results, we recommend that if applicable you deposit your laboratory protocols in protocols.io, where a protocol can be assigned its own identifier (DOI) such that it can be cited independently in the future. For instructions see: http://journals.plos.org/plosone/s/submission-guidelines#loc-laboratory-protocols

We look forward to receiving your revised manuscript.

Kind regards,

Helena Kuivaniemi, MD, PhD

Academic Editor

PLOS ONE

Journal Requirements:

2. Please ensure that you refer to Figures 4-7 in your text as, if accepted, production will need this reference to link the reader to the figure.

Reviewers' comments:

Reviewer's Responses to Questions

**Comments to the Author**

1. Is the manuscript technically sound, and do the data support the conclusions?

Reviewer #1: Partly

Reviewer #2: Yes

Reviewer #3: Partly

2. Has the statistical analysis been performed appropriately and rigorously? 

Reviewer #1: Yes

Reviewer #2: No

Reviewer #3: No

3. Have the authors made all data underlying the findings in their manuscript fully available?

Reviewer #1: Yes

Reviewer #2: Yes

Reviewer #3: Yes

4. Is the manuscript presented in an intelligible fashion and written in standard English?

Reviewer #1: Yes

Reviewer #2: Yes

Reviewer #3: Yes

5. Review Comments to the Author

Reviewer #1: The manuscript by Mwogi is a useful evaluation of the time involved in the generation of a routine lab report from time of test order through to result availability to clinicians. The comments below are minor, but deserve to be addressed by the authors

- This does not appear to have an author from the laboratory which seems inappropriate. If laboratory management was not involved in this study, that is an indictment of the difficulty that clinical laboratories often face. They are held to certain standards, but not always included in decision-making processes. The manuscript often has a negative tone that seems to impugn the efforts of the laboratory when in fact this type of evaluation is a necessary and useful part of laboratory quality improvement. One would hope that this was a collaborative and affirming research endeavor and the authors should mention this aspect in a revised manuscript

- The authors state on line 43 of page 3 that “timeliness… is the most crucial” aspect of laboratory performance. This is patently untrue as result accuracy is far more important to patient safety and management. Timeliness is AN important factor, but receiving inaccurate results quickly is useless. The authors should revise this statement.

- The meaning of lines 49-51 is unclear – please reword this section

- There is no description of the organization of the laboratory. Is this a 24/7 operation or is it day-shift only? Are assays run in batches or continuously? Is there a STAT process that has different turnaround characteristics? Does the laboratory have a lab information system (LIS)?

- The authors sometimes use “biochemistry” and other times use “electrolytes”. Please be consistent throughout the manuscript.

- The lab tests chosen are not often needed on a STAT basis. Thus the authors need to discuss the patient implications of receiving a result the same day versus the next day. If these test results are not needed urgently (and in fact is there anyone to receive results if they were issued earlier but in non-peak hours such as 1 am?).

- Were research assistants really working 24 hours continuously to follow samples and testing and results? This seems unlikely.

- It appears that the median is a better measure of central tendency here as there may be some substantial outliers. A histogram of TAT would be useful.

- Lines 169-171 seem contradictory. Please clarify

- Remove the redundant repetition of results from the discussion section. Instead focus on causes and potential solutions which would be helpful for readers to understand.

- The authors seem to be assigning blame to the lab when in fact, this is an excellent opportunity to champion the lab’s needs. Laboratories are often so tightly funded with administrators paying only for reagents and tech time while not taking many of the activities described here into account. Health care organizations need a call to arms to better support excellence in laboratory diagnostics. For example, at some institutions where an LIS is available and can link to an EMR, clinicians refuse to look up results preferring to wait for paper copies. The authors have clearly described dependence on paper reports as a bottleneck, but the solutions cannot come from the lab alone., The authors should engage is the effort of quality improvement by taking a stand on what is needed in order to improve lab services.

Reviewer #2: 1. Please provide a reference for the "Hawthorne effect".

2. Figure 1

Please consider modifying this diagram to include arrows indicating the flow and the role players at each step, i.e. doctors, nurses, porters, laboratory admin personnel, laboratory technicians etc. This will help a general reader to understand the flow practically. Also please indicate where each step takes place - the bedside, the nurses' or admin office in the ward, the receipt area in the laboratory, the analytical areas - again for a general reader who do not work in a hospital to understand the practical flow better.

3. Figure 3

Using 1 boxplot per time slot in this graph will convey more useful information to the reader, like the spread of TAT and where the central 50% lies. Since you state that you are reporting mean and SD according to a recommendation by Hawkins, you may consider displaying the mean and SD in these boxplots, instead of the more common median and IQR.

4. Table 3

Please clarify what comparison the p-value represents - is it Monday vs all the other days, or is it a Kruskal-Wallis test that includes all the days, in which case a significant p-value does not necessarily mean that "early" and "late" differ, but only that day of the week influences the rankings of TAT, giving no clue which day specifically has the greatest influence.

It is also not ideal to show mean and SD next to a p-value for a test that does not compare the means of groups. Reporting the medians and inter-quartile ranges would be more appropriate in combination with such a p-value. As above, I understand that you are following a specific recommendation for the TAT field in reporting the means - if you wish this table to remain consistent with that, then removing the p-values from both table 3 and 4 and creating an additional table for all comparisons with their p-values may be another way to present the p-values without them seeming to represent a difference in means.

I would suggest creating an "early" group Monday-Tuesday and a "late" group Wednesday-Thursday-Friday and just testing those with Kruskal-Wallis so that the p-value corresponds to the question you are interested in. Or trying log-transforming the data and then doing an ANOVA and post-hoc tests, to justify which days are different from the global mean.

It is also an option to do no statistical test, but show the TAT in a separate boxplot per day, so that the readers can see for themselves what the spread per day looks like and how days differ from each other. I do not think the absence of a p-value for the influence of day of the week diminishes your point that it is an important factor to consider when applying an analysis of where and why bottle necks occur.

5. Table 4

As before, it is not ideal to report mean and SD along with a p-value of a test that does not test a difference in means.

5. General comments

Overall, this paper is well written and clear.

Line numbers seem to have accidentally made their way into the text at places - please see lines 165 to 167 where the numbers "158" and "159" appear, as an example. Please check the rest of the text also for these artefacts.

Reviewer #3: Detailed comments on Manuscript number: PONE-D-19-21599

ABSTRACT:

1. Abstract format – does not follow the strict flow of the PLOS1 guidance. Authors must refer to the guidelines and apply them accordingly.

2. Use of SD in abstract – more useful statistical measures such as mean, range, CI and other more informative stats recommended

3. Bold statement made regarding significance but no objective test of outcome significance is mentioned

INTRODUCTION:

1. Generally verbose without communicating any additional or useful facts.

2. The quoting of reference 23 to support the preceding statement is inaccurate as the reference does not say what is stated. Either a better reference is found or the statement modified or removed.

3. The preferred “user” definition of TAT is acceptable with regards to “availability” of results, however, availability must include verbal, telephonic, electronic and social-media modalities of result communication. This omission implies that clinically critical results are not communicated by the foregoing methods. Is this the claim that the authors are making?

****This point must be attended to and clarified unequivocally as it affects the Discussion and other sections of this well-designed study***

4. The inclusion of reference 25 in the middle of a sentence must be rectified.

5. The last statement in the introduction regarding the fact that TAT studies are few in LMIC needs referencing as the reviewer’s impression is that there are such studies in the literature.

METHODS:

1. Under Setting, the number of technologists is stated but not other key staff such as Pathologists. It is important to make the distinction between a technologist vs pathologist-led laboratory.

2. The authors must be commended for an excellent, if labour intensive study design.

3. Note typing error UEC is Urea (not Urine) Electrolyte and Creatinine

4. Use of colloquial terms and abbreviations such as lab for laboratory, must be rectified.

RESULTS:

1. The use of SDs and medians and their contribution to the analysis is questionable. As this affects the overall impact of the paper, the authors are strongly urged to consult their statisticians and only include the most impactful measures. Reviewer recommends, mean, range, CI, 90th percentile, and medians. Consistency is lacking in the statistical measure (s) utilised.

2. Table 2 has a wealth of information, however, there are several errors in the figures stated in the discussion as they do not match those stated on Table 2.

3. Testing for the statistical significance or lack thereof, of the various differences observed is generally lacking. It is only done in the context of Biochemistry versus Haematology.

DISCUSSION:

1. Several mistakes and discrepancies noted between the results shown on Table 2 and those stated in the text. This should be easy to fix.

2. Authors should consider further defining “pre-analytical time”; “time from ordering to receipt in the laboratory”, “time from receipt in the laboratory to time of analysis” are examples of different ways of defining “pre-analytical”. In practical terms these times require different interventions to rectify. For instance time spent within the laboratory before specimen analysis is entirely within the control of the laboratory whereas time before reaching the laboratory is not. In general, “pre-analytical” phase refers to the whole time interval before the specimen is analysed and not as the authors narrowly define it, as “from arrival in the laboratory to start of analysis”

***This is another key area for revision and clarification.

3. The key factual findings that results were “out” in the laboratory but not accessible to the clinicians can only be true if it is confirmed that the telephone and electronic means of communicating results were not used. The authors must expressly state if this is the case.

4. The figure regarding the contribution of the “non-analytical” phase (96%) needs recalculation.

5. Time comparisons of TATs must only be made with similarly defined TATs in order to be meaningful.

6. Speculative aspects of the discussion must be minimised, for instance staff being tired as the week progresses.

7. The findings that TATs were higher with more staff available must be analysed in the context of a meaningful denominator such as number of samples per staff member. This cannot be a simple case of “too many cooks spoil the broth”

END

6. PLOS authors have the option to publish the peer review history of their article (what does this mean?). If published, this will include your full peer review and any attached files.

Reviewer #1: No

Reviewer #2: Yes: Elizna Maasdorp

Reviewer #3: No

---

## [Author Response · Author response to Decision Letter 0]

13 Feb 2020

RESPONSE TO REVIEWERS

Reviewer #1: The manuscript by Mwogi is a useful evaluation of the time involved in the generation of a routine lab report from time of test order through to result availability to clinicians. The comments below are minor, but deserve to be addressed by the authors

- This does not appear to have an author from the laboratory which seems inappropriate. 

The 4th author (RT) is actually the head of laboratory services division in the same institution. We have edited the manuscript to also acknowledge the critical work contributed by other laboratory managers in the institution towards this work. - PAGE 1

If laboratory management was not involved in this study, that is an indictment of the difficulty that clinical laboratories often face. They are held to certain standards, but not always included in decision-making processes. The manuscript often has a negative tone that seems to impugn the efforts of the laboratory when in fact this type of evaluation is a necessary and useful part of laboratory quality improvement. One would hope that this was a collaborative and affirming research endeavor and the authors should mention this aspect in a revised manuscript 

This work was partly supported by the same hospital and indeed the laboratory as part of its quality improvement efforts. The parts of the manuscript that may have inadvertently potrayed the laboratory and laboratory staff in bad light were unintentional and have been rectified. We received immense support from every aspect of the hospital in doing this work. Indeed every personnel involved in the chain from ordering of tests to availability of results was intimately involved in the planning and execution of the research. There was a general sense of the capability of the paper to point out weak points for improvement without necessarily apportioning blame. – Lines 262 -267

- The authors state on line 43 of page 3 that “timeliness… is the most crucial” aspect of laboratory performance. This is patently untrue as result accuracy is far more important to patient safety and management. Timeliness is AN important factor, but receiving inaccurate results quickly is useless. The authors should revise this statement. 

This was quoted from literature. We agree with the reviewer that indeed timeliness without accuracy is possibly even more harmful to patients. The paragraph has been edited to give equal importance to timeliness and accuracy – Lines 37 -39

- There is no description of the organization of the laboratory. Is this a 24/7 operation or is it day-shift only? Are assays run in batches or continuously? 

The setting subheading under methods has been updated and more detail has been added on the organization of the laboratory and a clarification on the specific aspect of the process that was evaluated. For example, STAT processes that co-exist were not evaluated (e.g. Random blood sugar tests). The hospital does not use and laboratory information system. All processes are paper based except billing – Lines 68 -73

- The authors sometimes use “biochemistry” and other times use “electrolytes”. Please be consistent throughout the manuscript.

We have standardized the use of terms in the manuscript for consistency. 

- The lab tests chosen are not often needed on a STAT basis. Thus the authors need to discuss the patient implications of receiving a result the same day versus the next day. If these test results are not needed urgently (and in fact is there anyone to receive results if they were issued earlier but in non-peak hours such as 1 am?). 

We have updated the discussion and addressed the question of ‘how urgent do these results need to be delivered?’. We indeed agree with the reviewer on this valid question. However, while it may not be necessary to deliver laboratory results STAT, unnecessary delay is of patient safety concern. Additionally, A 2001 Q-Probes study concluded that the optimal time from order to reporting for biochemistry tests was 47 minutes while that for hematology was 35 minutes. Any delay to get the results back to the clinicians longer than these is considered suboptimal – Lines 181 -183

.

- Were research assistants really working 24 hours continuously to follow samples and testing and results? This seems unlikely. 

RAs followed the ordering and collection of samples during normal working hours and collected lab related information from equipment time stamps. The RAs also 90 collected relevant lab time-stamps from the laboratory computers system, as time-stamps were generated when the laboratory test was both analyzed and when the results were printed.The research assistants were not working 24 hours. This was not necessary given the setting of the inpatient ward in the hospital. All laboratory tests were ordered during ward rounds in the morning. The phlebotomists would then do sample collection late mornings and deliver all samples to the laboratories by afternoons. Processing and printing of results as well as dispatch was done in a batched manner. There was largely no inpatient laboratory processing activities in the evenings and during the nights. – Lines 80 -82

- It appears that the median is a better measure of central tendency here as there may be some In total agreement with the reviewer. We have changed the tables to boxplots to better reflect the spread of the TAT with more emphasis on the median. – Fig 3 - 5

-Lines 169-171 seem contradictory. Please clarify

We have updated the manuscript to eliminate the contradictory and confusing manner of the two statements. Using the Kruskal Wallis to test for significant difference between the means, we could not find an association between the number of personnel present and the TAT. However, it was noted that in some instances (Number of laboratory personnel for example), the more personnel present, the longer the TAT tended to be. – Lines 168 - 170

- Remove the redundant repetition of results from the discussion section. Instead focus on causes and potential solutions which would be helpful for readers to understand.

This was well noted. The discussion has been reformatted to remove any unnecessary inclusion of items already included in the results. – Lines 172 onwards

- The authors seem to be assigning blame to the lab when in fact, this is an excellent opportunity to champion the lab’s needs. Laboratories are often so tightly funded with administrators paying only for reagents and tech time while not taking many of the activities described here into account. Health care organizations need a call to arms to better support excellence in laboratory diagnostics. For example, at some institutions where an LIS is available and can link to an EMR, clinicians refuse to look up results preferring to wait for paper copies. The authors have clearly described dependence on paper reports as a bottleneck, but the solutions cannot come from the lab alone., The authors should engage is the effort of quality improvement by taking a stand on what is needed in order to improve lab services.

We noted this crucial observation. The paper may have unintentionally set the wrong tone. However, we indeed found that the laboratories process the tests in the shortest time possible (almost close to the acceptable standards). However, there are systemic issues beyond just the laboratory e.g. number of personnel available may not be adequate to deliver individual laboratory results. Instead they are batched and delivered. Clinicians mark all laboratory test orders as ‘urgent’ and contribute to delaying all results as none emerges as urgent to care for example. Generally, what emerged is that there needs to be collaborative efforts as well as systemic changes to improve on the TAT. The biggest changes to improve the TAT is on the pre-analytical and post-analytical stage and not necessarily the laboratory itself.

 

Reviewer #2: 1. Please provide a reference for the "Hawthorne effect".

A reference has been added – Reference 22 Line 86

2. Figure 1

Please consider modifying this diagram to include arrows indicating the flow and the role players at each step, i.e. doctors, nurses, porters, laboratory admin personnel, laboratory technicians etc. This will help a general reader to understand the flow practically. Also please indicate where each step takes place - the bedside, the nurses' or admin office in the ward, the receipt area in the laboratory, the analytical areas - again for a general reader who do not work in a hospital to understand the practical flow better.

The diagram has been modified appropriately as suggested. Arrows have been added to indicate the flow linking pre-analytical to analytical to post-analytical. The places where these activities take place have also been indicated in the diagram. – Figure 1

3. Figure 3

Using 1 boxplot per time slot in this graph will convey more useful information to the reader, like the spread of TAT and where the central 50% lies. Since you state that you are reporting mean and SD according to a recommendation by Hawkins, you may consider displaying the mean and SD in these boxplots, instead of the more common median and IQR.

Figure 3 has been updated appropriately according to the reviewers recommendation. For each timeslot, boxplot has been used instead. – Figure 3 to 5

4. Table 3

Please clarify what comparison the p-value represents - is it Monday vs all the other days, or is it a Kruskal-Wallis test that includes all the days, in which case a significant p-value does not necessarily mean that "early" and "late" differ, but only that day of the week influences the rankings of TAT, giving no clue which day specifically has the greatest influence.

It is also not ideal to show mean and SD next to a p-value for a test that does not compare the means of groups. Reporting the medians and inter-quartile ranges would be more appropriate in combination with such a p-value. As above, I understand that you are following a specific recommendation for the TAT field in reporting the means - if you wish this table to remain consistent with that, then removing the p-values from both table 3 and 4 and creating an additional table for all comparisons with their p-values may be another way to present the p-values without them seeming to represent a difference in means.

I would suggest creating an "early" group Monday-Tuesday and a "late" group Wednesday-Thursday-Friday and just testing those with Kruskal-Wallis so that the p-value corresponds to the question you are interested in. Or trying log-transforming the data and then doing an ANOVA and post-hoc tests, to justify which days are different from the global mean.

It is also an option to do no statistical test, but show the TAT in a separate boxplot per day, so that the readers can see for themselves what the spread per day looks like and how days differ from each other. I do not think the absence of a p-value for the influence of day of the week diminishes your point that it is an important factor to consider when applying an analysis of where and why bottle necks occur.

5. Table 4

As before, it is not ideal to report mean and SD along with a p-value of a test that does not test a difference in means.

After further discussion with the biostatistician, we agreed with the reviewers last point and we have removed table 3 and table 4 and replaced it with a boxplot that better depicts the spread of the TAT per day and the spread of the TAT for the number of personnel. – Fig 3 - 5

5. General comments

Overall, this paper is well written and clear.

Line numbers seem to have accidentally made their way into the text at places - please see lines 165 to 167 where the numbers "158" and "159" appear, as an example. Please check the rest of the text also for these artefacts.

A further review of the whole manuscript was further done to remove the mixup where line numbers got mixed up in the text during formatting. 

 

Reviewer #3: Detailed comments on Manuscript number: PONE-D-19-21599

ABSTRACT:

1. Abstract format – does not follow the strict flow of the PLOS1 guidance. Authors must refer to the guidelines and apply them accordingly.

Latex template was used and during the conversion from latex to PDF and then to word, the formatting was lost. We have attached a PDF as reference to the structure of the document. We have also rectified the formatting to reflect Plos one guidelines.

2. Use of SD in abstract – more useful statistical measures such as mean, range, CI and other more informative stats recommended

We have changed the statistical measures to use of 95% confidence intervals. This has been standardized throughout the document. – Abstract and Lines 148 - 153

3. Bold statement made regarding significance but no objective test of outcome significance is mentioned

We have removed speculative statements in the document and in the abstract that are not backed by data.

INTRODUCTION:

1. Generally verbose without communicating any additional or useful facts.

We felt that the introduction was crucial in painting the necessary background to justify why a time motion study would be appropriate in understanding workflow processes that influence TAT. We have edited the introduction for clarity and some paragraphs have been removed altogether.

2. The quoting of reference 23 to support the preceding statement is inaccurate as the reference does not say what is stated. Either a better reference is found or the statement modified or removed.

The reference has been updated to the correct one. Ref 19

3. The preferred “user” definition of TAT is acceptable with regards to “availability” of results, however, availability must include verbal, telephonic, electronic and social-media modalities of result communication. This omission implies that clinically critical results are not communicated by the foregoing methods. Is this the claim that the authors are making?

****This point must be attended to and clarified unequivocally as it affects the Discussion and other sections of this well-designed study***

We have edited the methods to explicitly state that we only tracked printed/paper based results. We have added as a limitation of this study that if there were critical results that were communicated via other means, the results may overestimate the TAT. However, from the experience of the authors, very few critical results are communicated. A follow up paper that followed only critical results established this. Lines 66 -73

4. The inclusion of reference 25 in the middle of a sentence must be rectified.

This has been rectified.

5. The last statement in the introduction regarding the fact that TAT studies are few in LMIC needs referencing as the reviewer’s impression is that there are such studies in the literature.

This statement has been removed as it may be deemed speculative.

METHODS:

1. Under Setting, the number of technologists is stated but not other key staff such as Pathologists. It is important to make the distinction between a technologist vs pathologist-led laboratory. 

The section has been edited to include other relevant personnel. The management team is led by a technologist with 6 pathologists serving as consultants in the histology laboratory (1 full time and 5 part time) – Lines 71 - 73

2. The authors must be commended for an excellent, if labour intensive study design.

3. Note typing error UEC is Urea (not Urine) Electrolyte and Creatinine

This has been rectified.

4. Use of colloquial terms and abbreviations such as lab for laboratory, must be rectified.

This has been rectified.

RESULTS:

1. The use of SDs and medians and their contribution to the analysis is questionable. As this affects the overall impact of the paper, the authors are strongly urged to consult their statisticians and only include the most impactful measures. Reviewer recommends, mean, range, CI, 90th percentile, and medians. Consistency is lacking in the statistical measure (s) utilised.

The results section has been reworked and for consistency, we have reported the means at 95% confidence intervals. The table 2 also include the 90th Percentile and the median. – Abstract, Results and Discussion section

2. Table 2 has a wealth of information, however, there are several errors in the figures stated in the discussion as they do not match those stated on Table 2.

We have rectified the inconsistencies and matched the figures stated in Table 2 with those in the results section. - Table 2, Results and Discussion

3. Testing for the statistical significance or lack thereof, of the various differences observed is generally lacking. It is only done in the context of Biochemistry versus Haematology.

The calculation of the sample size may not have been powered high enough to detect various differences. We have adopted another reviewer suggestion that we include boxplots that will help readers quickly discern relationships between the various parameters. We have removed Table 3 and 4 and replaced them with boxplots. – Figures 3 - 5

DISCUSSION:

1. Several mistakes and discrepancies noted between the results shown on Table 2 and those stated in the text. This should be easy to fix.

This has been fixed and together with recommendation from another reviewer, we have removed repetitions of results in the discussion section.

2. Authors should consider further defining “pre-analytical time”; “time from ordering to receipt in the laboratory”, “time from receipt in the laboratory to time of analysis” are examples of different ways of defining “pre-analytical”. In practical terms these times require different interventions to rectify. For instance time spent within the laboratory before specimen analysis is entirely within the control of the laboratory whereas time before reaching the laboratory is not. In general, “pre-analytical” phase refers to the whole time interval before the specimen is analysed and not as the authors narrowly define it, as “from arrival in the laboratory to start of analysis”

***This is another key area for revision and clarification.

The methods section has been updated to show the adopted definition based on Lundbergs definition of therapeutic TAT. The pre-analytic TAT was defined as from the point of order of tests to the point of receipt in the laboratory for processing. – Lines 75 - 78

3. The key factual findings that results were “out” in the laboratory but not accessible to the clinicians can only be true if it is confirmed that the telephone and electronic means of communicating results were not used. The authors must expressly state if this is the case.

We have added this as a limitation as we did not track the telephone communications for critical results in this study. Very few critical results are communicated and there is a separate paper that specifically deals with communication of critical results that demonstrated this. However, there are no electronic means of communications of results. – Lines 226 - 235

4. The figure regarding the contribution of the “non-analytical” phase (96%) needs recalculation.

We have re-worded this paragraph so that it is now more clear. The meaning of this is that pre-analytical and post-analytica delays contribute up to 96% prolongation of the therapeutic TAT and indeed the laboratory delays are minimal. – Lines 43 - 46

5. Time comparisons of TATs must only be made with similarly defined TATs in order to be meaningful.

We have stated this as a major challenge that affects all studies that try to do comparisons of TAT as studies differ in the definition of TAT. A standardization of TAT is needed to improve on the comparisons.

6. Speculative aspects of the discussion must be minimised, for instance staff being tired as the week progresses. 

We have removed the speculative aspects of the discussion and limited the discussion to only the available results. 

7. The findings that TATs were higher with more staff available must be analysed in the context of a meaningful denominator such as number of samples per staff member. This cannot be a simple case of “too many cooks spoil the broth” 

We have added a boxplot with the number of samples for each number of staff included in order to aid in interpretation. We have also stated that this is an interesting finding that may require further exploration as we may not have all the data to explain this phenomenon. – Figure 5, Figure 6

---

## [Decision Letter · Decision Letter 1]

11 Mar 2020

Therapeutic turnaround times for common laboratory tests in a tertiary hospital in Kenya

PONE-D-19-21599R1

Dear Dr. Mwogi,

We are pleased to inform you that your manuscript has been judged scientifically suitable for publication and will be formally accepted for publication once it complies with all outstanding technical requirements.

Congratulations!

With kind regards,

Helena Kuivaniemi, MD, PhD

Academic Editor

PLOS ONE

Additional Editor Comments (optional):

Reviewers' comments:

Reviewer's Responses to Questions

**Comments to the Author**

1. If the authors have adequately addressed your comments raised in a previous round of review and you feel that this manuscript is now acceptable for publication, you may indicate that here to bypass the “Comments to the Author” section, enter your conflict of interest statement in the “Confidential to Editor” section, and submit your "Accept" recommendation.

Reviewer #1: All comments have been addressed

Reviewer #2: All comments have been addressed

Reviewer #3: All comments have been addressed

2. Is the manuscript technically sound, and do the data support the conclusions?

Reviewer #1: Yes

Reviewer #2: (No Response)

Reviewer #3: Yes

3. Has the statistical analysis been performed appropriately and rigorously? 

Reviewer #1: N/A

Reviewer #2: (No Response)

Reviewer #3: Yes

4. Have the authors made all data underlying the findings in their manuscript fully available?

Reviewer #1: Yes

Reviewer #2: (No Response)

Reviewer #3: Yes

5. Is the manuscript presented in an intelligible fashion and written in standard English?

Reviewer #1: Yes

Reviewer #2: (No Response)

Reviewer #3: Yes

6. Review Comments to the Author

Reviewer #1: (No Response)

Reviewer #2: (No Response)

Reviewer #3: All the my comments and suggestions have been adequately addressed.

Congratulations and well done.

7. PLOS authors have the option to publish the peer review history of their article (what does this mean?). If published, this will include your full peer review and any attached files.

Reviewer #1: No

Reviewer #2: Yes: Dr Elizna Maasdorp

Reviewer #3: Yes: Zivanai Cuthbert Chapanduka

---

## [Editor Report · Acceptance letter]

18 Mar 2020

PONE-D-19-21599R1 

Therapeutic turnaround times for common laboratory tests in a tertiary hospital in Kenya 

Dear Dr. Mwogi:

I am pleased to inform you that your manuscript has been deemed suitable for publication in PLOS ONE. Congratulations! Your manuscript is now with our production department. 

With kind regards,

on behalf of

Professor Helena Kuivaniemi 

Academic Editor

PLOS ONE